# Ecolodge Entrepreneurship in Emerging Markets: A New Typology of Entrepreneurs; The Case of IRAN

**Hojjat Varmazyari** [1,*] , **Seyed Hamid Mirhadi** [1], **Marion Joppe** [2] , **Khalil Kalantari** [1] **and Alain Decrop** [3]

1 Department of Agricultural Management and Development, Faculty of Agricultural Economics and Development, University of Tehran, Karaj 31587-77871, Iran; mirhadi@ut.ac.ir (S.H.M.); khkalan@ut.ac.ir (K.K.)
2 School of Hospitality, Food and Tourism Management, University of Guelph, Guelph, ON N1G 2W1, Canada; mjoppe@uoguelph.ca
3 Faculty of Economics, Social Sciences and Business Administration, University of Namur, 5000 Namur, Belgium; alain.decrop@unamur.be
* Correspondence: varmazyari@ut.ac.ir; Tel.: +98-912-4966174; Fax: +98-261-2238293

**Abstract:** This study aims to clarify how ecolodge entrepreneurship evolves from idea formation to ecolodge establishment in emerging markets. The related process affects authentic ecolodge development. The research employed grounded theory to explore this process and its implications to examine for the first time how individuals enter the ecolodge industry in an emerging market. The interaction of four constructs (namely drivers, motives, context, and idea sources) explains the costs and benefits that ecolodge entrepreneurs perceive in entering this industry. Moreover, we develop a new typology of tourism entrepreneurs in an ecolodge context based on the combined approach. Entrepreneurs are classified into three segments, including ecolodge lovers, cool job seekers, and young detached entrepreneurs. Although the ecolodge lovers were most in line with the principles of sustainable tourism and most likely to set up authentic ecolodges, most of the entrepreneurs belonged to the other two clusters. The explored process and typology highlight coordinated action in the development of ecolodges.

**Keywords:** ecolodge; entrepreneurship; typology; rural development

## 1. Introduction

Since the mid-1960s, the prevalence of up–bottom planning approaches and economic profit-oriented systems in the tourism sector led to increase in the demand for restrictions in its development [1]. By the late 1980s, the school of "alternative tourism", such as ecotourism, was firmly instituted, and by the mid-1990s, sustainable tourism development was acknowledged as "the new [tourism] industry paradigm" [2]. Sustainable tourism development in rural areas is one of the most important strategies for the diversification of jobs and eradication of poverty. Due to increasing global industrialization and urbanization, reinforced by the COVID-19 pandemic [3], the demand for spending leisure time in the rural and natural environment has increased [4–6]. The Organization for Economic Co-operation and Development [7] predicted that evolving visitor demand and sustainable tourism growth are two of four megatrends that will affect tourism in the 2040 horizon.

Different hotel concepts emerged in parallel to new travel trends, such as ecohotels, ecolodges, and green hotels [8]. Ecolodges are one of the most common options for satisfying the new tourist tendencies since they provide accommodation and services to tourists seeking a responsible use of nature by learning from it and wishing to contribute to the well-being of the local community. Indeed, the ecolodge industry can have a significant impact on the livelihood and culture of villagers and endogenous tribes due to its location in natural and rural areas. In addition, it can also help with conservation efforts and enhance tourists' enjoyment of nature [9].

The development of an ecolodge as an ecotourism enterprise is a difficult undertaking because, in theory, the entrepreneur operates on a market basis while partnering fairly and continuously with the local community and also protecting the environment [10]. Ecotourism occurs in natural areas, conserves the environment, includes environmental education, and is managed to be sustainable [11]. Relatively few tourism and hospitality businesses show a holistic concern for sustainability, and the more conventional business models have been challenged [12,13]. The term 'ecotourism' has been misused by numerous actors as a convenient label. There are contesting definitions of ecotourism, ecotourists, and ecolodges, so the expansive and divergent applications of these terms have made them all but meaningless [14–18]. Due to their rich cultural heritage and many natural attractions, emerging countries welcome any model of ecotourism development regardless of its long-term sociocultural and ecological impacts [19,20] to respond to the growing demand for outdoor recreation. For example, Iran, as an emerging country, has taken steps to develop this industry in line with global trends and national development goals, particularly for rural areas. The number of Iranian ecolodges has experienced a 4.7-fold increase, from 320 in 2017 to 1500 units in 2020.

However, there are growing concerns that this rapid expansion could challenge the industry's sustainability. It is also critiqued for not having inclusive positive impacts on the environment and the development of the local community [9,14], while the overall contribution of ecotourism to sustainable development is considered ambiguous [21]. Pawliczek and Mehta [22] noted the same issues in Madagascar, where "lodges calling themselves 'ecolodges' are mushrooming [ . . . ] as a lot of traditional lodge owners use the term for promotional activities without a proper understanding of the value of the term". Traditional business wisdom places the greatest emphasis on economic profitability and return on investment, but this mindset is dangerous for the sustainability and protection of the environment and the rights of the local community [10,17] and confirms that local community members often enter ecotourism business without enough tourism knowledge and experience [23].

Although many scholars have expressed reservations about ecotourism development [19,21,23–30], these concerns have not been supported by in-depth studies on the motives and process of entrepreneurs' entry into the ecolodge industry. Indeed, there has been no empirical analysis to date of how entrepreneurs choose to enter ecolodge businesses in emerging markets. Understanding their motives and the process is critical to determining whether the expected positive economic, social, and environmental outcomes are achieved. In emerging markets such as Iran, where the industry is at an early stage of development and where hierarchical top–down and sectoral administrative approaches are dominant [31], these insights can improve both policy and entrepreneurial decision-making. The present study therefore set out to answer the following questions:

- What motivated entrepreneurs to enter the ecolodge industry in emerging markets?
- What decision-making process did these entrepreneurs go through from idea formation to ecolodge establishment in emerging markets?
- What factors influenced their decision, and how did they perceive the impacts of ecolodge entrepreneurship in emerging markets?
- In these markets, can a typology of the entrepreneurs be developed based on the entrepreneurship process and ecolodge administration and services?
- What implications does this ideation and typology have for the development of authentic ecolodges in emerging markets?

This study first deals with the definition and requirements of an ecolodge and then analyzes the empirical literature on ecolodge and tourism entrepreneurship. In the following sections, the research method, findings, and the theoretical and practical achievements of this research have been explained.

## 2. Literature Review

The term "ecolodge" was first used in 1994 at Maho Bay Camps in the US Virgin Islands [9,32]. Later that year, The International Ecotourism Society (TIES) defined the term as an "industry label used to identify a nature-dependent tourist lodge that meets the philosophy of ecotourism" [33]. An ecolodge's underlying philosophy should stress the natural and cultural attractions, the educational and participatory experiences offered, and the manner and extent of involvement of the local populations. Above all, ecological sensitivity must define an ecolodge, including its integration into its surroundings to minimize negative environmental impacts [34].

### 2.1. Requirements of Ecolodges

Mehta [9] argued that an ecolodge is not merely a traditional hut, and to be truly eco-friendly, it should attract tourists seeking to understand nature. From his point of view, an ecolodge benefits the local community and engages them and provides tourists with a collaborative and active experience. He believed that an ecolodge should provide guests with an unparalleled experience and educate them by connecting them with nature, and it should be planned, designed, constructed, and managed in such a way that the least harm is done to nature and the local community [9,35]. Ceballos-Lascurain [34] went a step further and called for a new approach to architecture based on a structure's integration with the surrounding ecosystems, called 'ecodesign'.

According to the requirements and rules of ecotourism enterprises, different schemes, plans, and eco-labels have been developed in the world [36]. Indeed, eco-labels were commenced around 1978 to promote the best practices in the tourism sector and hinder practices such as water overuse, selling wildlife souvenirs, farmland use change, etc. [37]. Examples of 13 tourism eco-labels that are mainly established and managed by non-profit and non-governmental organizations in developed countries can be found in Table 1. These eco-labels are the best known that are exclusively applicable to tourism. Unfortunately, these initiatives are often underdeveloped or absent in emerging markets, where ecolodges have rarely registered for tourism certification.

**Table 1.** Best known tourism eco-labels.

| Code | Name | Logo | Countries | Year | Managing Organization |
|------|------|------|-----------|------|-----------------------|
| 1 | ECO certification |  | Malta | 2002 | Malta Tourism Authority |
| 2 | EcoLabel Luxembourg |  | Luxembourg | 1999 | Oekozenter Pafendall |
| 3 | Green Certificate |  | Latvia | 2001 | Latvian Country Tourism association |
| 4 | Green Tourism Business Scheme |  | Canada, Ireland, United Kingdom | 1997 | Green Business UK Ltd. |
| 5 | ibex fairstay |  | Switzerland | 2002 | ibex fairstay |

**Table 1.** *Cont.*

| Code | Name | Logo | Countries | Year | Managing Organization |
|------|------|------|-----------|------|-----------------------|
| 6 | International Eco Certification Program | | Australia | 1991 | Ecotourism Australia |
| 7 | Legambiente Turismo | | Italy | 1997 | Legambiente Turismo |
| 8 | Nature's Best Ecotourism | | Sweden | 2002 | Swedish Ecotourism Society |
| 9 | Sustainable Tourism Education Program (STEP) | | Ireland, Malawi, Mexico, United Kingdom, United States | 2007 | Sustainable Travel International |
| 10 | Viabono | | Germany | N/A | Viabono |
| 11 | Estonian Ecotourism Quality Label | | Estonia | N/A | Estonian Ecotourism Association |
| 12 | Calidad Galapagos | | Ecuador | N/A | Capturgal |
| 13 | Green Key | | Belgium, Canada, Cyprus, Denmark, Estonia, France, Greece, Italy, Japan, Jordan, Latvia, Lithuania, Morocco, Netherlands, Poland, Portugal, Russian Federation, Sweden, Tunisia, Ukraine | 1994 | Foundation for Environmental Education (FEE) |

For more information about these eco-labels refer to: www.ecolabelindex.com/ecolabels/ (accessed on 2 July 2022).

### 2.2. Empirical Research on Ecolodge and Tourism Entrepreneurship

Four types of empirical studies were reviewed for this study. The first group includes studies where experimental work in the field of ecolodges was undertaken. Scopus and Google Scholar databases were searched for the words "ecolodge" and "eco-lodge" until 2020, without any restrictions on the country, language, and worldview and approach of the studies. Merely article journal with empirical data were selected, i.e., chapters in books, book series, books, review articles, and conference proceeding were all excluded.

According to the narrative literature review, five themes have dominated: (1) unsustainable ecotourism business practices; (2) profiles of ecotourists, their motivations, characteristics, attitudes, satisfaction, and related segmentation; (3) service quality, attributes, performance goals, and sustainable management of ecolodges; (4) contribution of

ecolodges in local sustainable development; and (5) miscellaneous themes, namely private tourism ecolodge concessions and pedagogical processes for designing an ecolodge. Indeed, the process of entrepreneurs' entry into the ecolodge industry has been neglected. The relevant studies under each heading can be found in the Appendix A. These studies have been analyzed by the authors, main objective, sample, method of analysis, and key findings (Appendix A).

The second group of studies enumerated the factors that make up the macro- and micro-environment of tourism entrepreneurship, such as decreased government support programs for agriculture, increased agricultural production cost, weakness in agricultural trade laws, on-farm income and economic fluctuations, low income elasticity in the agricultural market, over-reliance on raw product sales, industrialization, encroachment of urban areas into rural ones, environmental pressures, and climate change, on one hand, and regional economics, information and communication technologies, human and social capital, financial services, appropriate entrepreneurial environment, and tax incentives on the other hand [36,38–44].

The third group of studies examined the motives of people who enter tourism in rural settings. These studies revealed motives such as responding to market needs, earning more, long-term financial security, improving job security, family pursuits, job creation for family members, decreasing anxiety and dissatisfaction with a previous job, improving quality of personal or family life, retirement, making better use of leisure time, having extra room, making full use of resources, owning a business, managing human resources, pursuing a personal interest, achieving independence, enjoying and entertaining, engaging with guests, talking to people, and educating tourists [36,45–50]. Amanor-Boadu [51] also presented variables, such as demographic characteristics, personal satisfaction, and economic and social motives, as determining the likelihood of farmers moving from one stage to another through their proposed three-step sequential framework for making farm diversification decisions. Representing a different viewpoint, Moraru et al. [52] and Set et al. [50] investigated the motives from the perspective of push and pull factors. Motives examined in previous studies can generally be divided into economic and social ones, with economic motives having played key roles in agricultural tourism in North America and the UK [49] in contrast to Australia, where social motives have been of greater importance. Ainley and Kline [38] attributed this to the positivist worldview of the study authors and their quantitative approaches, which did not consider the complexity and variety of factors affecting individuals' motives.

The fourth group of studies deals with the typology of tourism entrepreneurs. The previous studies on tourism entrepreneurship have mainly segmented tourism entrepreneurs into two clusters of growth-oriented and tourism lifestyle entrepreneurs [53–60]. Growth-oriented entrepreneurs have the self-confidence to administer a business, have a high propensity for risk-taking, and understand the value proposition. Their aim is to foster businesses that can compete, develop, and create employment. The economic benefits of the tourism business are a priority for these entrepreneurs [58]. Tourism lifestyle entrepreneurs, on the other hand, are motivated by non-economic motives [60], such as improving their quality of life, social networking, and strengthening cohesion and integrity of the local community [53,54,61–68]. They are innovative entrepreneurs who develop new products and may even have migrated to the destination just to start a tourism business [54,57,66]. A high place attachment amongst tourism lifestyle entrepreneurs is the source of local knowledge that, in turn, gives unique and different meaning to the experiences and services offered to tourists, creates differentiation, and increases the competitiveness and sustainability of the business model [56,69]. Dias et al. [56] revealed that entrepreneurial communication and local knowledge assimilation have a significant effect on the innovativeness and self-efficacy of tourism lifestyle entrepreneurs.

## 3. Materials and Methods

Based on the constructivist worldview, the present study employed a grounded theory strategy to explore entrepreneurs' motives and decision-making processes, from idea formation to start-up of the ecolodges, in Mazandaran Province, Iran. Grounded theory has been recommended for the "examination of subjectivity of experience" and for understanding the phenomenon from "research participants' point of view" [70], making it ideal for the current study, since the process of ecolodge entrepreneurship is unknown, as are its implications for authentic ecolodge development, particularly in emerging markets. The components of the business environment in ecolodge entrepreneurship and their impact on the sustainable development of ecolodge entrepreneurship are not known. Thus, it was necessary to conduct in-depth qualitative research to reveal the latent dimensions of ecolodge entrepreneurship for policy makers and, as a result, to make better decisions to form a more sustainable ecolodge entrepreneurship. Mazandaran Province is located on the south shore of the world's largest lake, the Caspian Sea, and it shares frontiers with four neighboring countries. This province is located north of Tehran, the capital of Iran. This region was chosen because it is the premier tourism destination in Iran, known for its many natural and cultural tourist attractions. Boasting forests, caves, waterfalls, rivers, numerous hot and cold mineral waters, springs, and wetlands, Damavand Peak, the highest mountain in Iran and the tallest volcano in Asia and the Middle East, is located here. Moreover, the province is located on the south shore of the Caspian Sea, the largest lake in the world. The province of Mazandaran has been the second province of Iran with 4.7 million overnight trips [71].

### 3.1. Selection of Interviewees

The interviewees in this study were selected according to theoretical sampling [72]. Entrepreneurs from 12 Mazandaran Province ecolodges, as the research focus, were selected using purposive sampling, where new interviewees were recognized throughout the data collection and coding process. In this study, entrepreneurs were individuals who had set up an officially approved ecolodge and generally owned it at the same time. The choice of the "eco" by entrepreneurs was not necessarily conscious, and the government's incentive and top-down policies were determinant in choosing this kind of "lodge". These accommodations tended to provide B&B services, usually in old traditional rural houses. Based on maximum variation sampling, the interviewees were varied as far as possible with the inclusion of extreme and atypical cases. Among the interviewees were entrepreneurs who had a college education but also some with very limited ability to read and write. Ecolodges near the main road were visited, as well as some that were more than two-hours'-drive into the forest from the city. Owners who had severe financial problems were interviewed, as were individuals who had strong financial backing and received a significant loan from the bank. In the stage of developing typology of the entrepreneurs, we returned to entrepreneurs with different business characteristics and services to collect the data necessary to saturate the categories and assure representativeness of the concepts. Entrepreneurs were interviewed at their ecolodges.

To improve the confirmability and transferability of research findings, based on a triangulation strategy, experts from the Cultural Heritage, Handicrafts, and Tourism Organization (CHTO) of Mazandaran and the villagers present in the visited rural areas were also interviewed. Saturation was achieved after 31 interviews were conducted with 15 owners of the ecolodges, 14 villagers, and two experts from CHTO. None of the owners had a related academic education or vocational training in tourism and hospitality and held different occupations prior to establishing the ecolodge.

### 3.2. Data Collection and Analysis

Data were collected through semi-structured interviews, and data gathering tools were note taking and recordings of interviews. Interviews were conducted individually to explore the depth of the issues and encourage interviewees to speak freely. Each interview

took an average of one hour. By presenting a business card with academic credentials and affiliation, the lead researcher assured the interviewees that the research was conducted solely for the purpose of discovering the concepts for academic purposes and had no administrative and regulatory function. Interviews were conducted by a senior student of a master's degree in rural development who had completed a two-unit course in these subjects and had studied the literature under the supervision of two faculty members specialized in agritourism and rural tourism. Leading questions and comments were avoided to elicit the interviewee's own viewpoints and experiences. Unclear conceptions and uncertain understandings of the interviewer were scrutinized through feedback from interviewees.

In addition, the two experts from CHTO of Mazandaran were asked to explain the generalities of ecolodges and the related steps. Fourteen villagers in the rural areas surveyed were asked about their views on ecolodges and the reasons for not entering the business. Data analysis was undertaken simultaneously with data collection in three stages of open coding, axial coding, and selective coding [73]. The iterative process of the grounded theory allowed for continuous improvement of the interviews and questioning, adding new questions, and deepening of the discussions. One of the faculty members independently undertook the coding to verify that the student was consistent. During the research, the interviewer discussed the findings with the faculty members in a triangulation process and applied the necessary modifications to the ongoing interviews and coding. The interviews were analyzed by focusing on what the interviewees stated and avoiding imposition of the researchers' structures and assumptions as far as possible. The codes from the different interviews were compared and placed in different categories. In the next step, the necessary interpretations were made to establish connections between the concepts discovered and to elaborate a more global theoretical framework. By employing inductive reasoning, this research moved from observations and data to theory construction.

## 4. Findings

In-depth analysis of the interviews provided unique and comprehensive insights into the process of entrepreneurs' entry into the ecolodge industry in rural areas. Iterative analysis and comparisons categorized the findings into four distinct categories, including entrepreneurs' drivers, motives, idea sources, and contexts.

### 4.1. Drivers

Those involved in the ecolodge business recounted a set of circumstances or events that had stimulated them to enter the business. These situations or events have been categorized as drivers, factors that shape and reinforce one's motive. Six drivers are described below.

### 4.1.1. Poor Working Conditions

Prior to entering the business, some interviewees had made efforts to be recruited by government departments or private corporations. However, they had failed to do so due to the harsh conditions of government employment, low salaries, and one-sided and inequitable private sector contracts. One of the young interviewees with tertiary-level education said:

> "I went to several government departments for employment. Sometimes government jobs needed favoritism. Private companies pay too low salaries and just want to sign a one-sided contract right from the start."

Another interviewee, a 40-year-old man with primary education, described his previous job:

> "I was working in a construction office in Tehran for a while. My boss had many irrelevant demands. For example, my work time was finished at 14 o'clock. He said you should sit here until 5pm. You're not allowed to go. [ . . . ] I couldn't stand it anymore. I talked back and left."

These inappropriate working conditions prompted both men to find a better job and become self-employed.

### 4.1.2. Economic Failure

Some interviewees had experienced economic failure and bankruptcy in the not-too-distant past. Because they previously had decent jobs and lived in the city, they had a relatively luxurious lifestyle compared to the rural population, and this economic failure forced them to pursue a new activity due to the difficult financial conditions. For example, one interviewee, who was previously a green space contractor of the Municipality of Tonekabon, said:

*"I used to be a floriculturist and green space contractor. I also had a plant nursery. But they didn't pay me, and I decided to start a new job."*

Another interviewee had been a shopkeeper in Behshahr before entering the ecolodge business. He believed that his grocery store was the largest in the city and went bankrupt for personal reasons.

### 4.1.3. Unfavorable Conditions for Livelihood Provision

Some interviewees described their predicament as a difficult economic situation that made them look for a new job. Lack of some living amenities, dependency on parents to provide for their living expenses, and heavy debt were some of the hardships. For example, one of the interviewees described her family difficulties:

*"My husband is a worker. We had to do something. Otherwise, our home construction would not be completed with my husband's previous earnings, i.e., wage labor. [ . . . ] My husband did whatever he could, and spent his income on this house's construction . . . "*

### 4.1.4. Childhood Influences

One of the interviewees mentioned how her husband's childhood interests in wooden huts incited him to establish an ecolodge:

*"As a child, he loved wooden huts and even occasionally built small and miniature huts on his father's farm."*

A female manager of an eco-lodge referred to her father's hospitality history as a driver for her entering the ecolodge entrepreneurship:

*"My father used to rent a room in our house to tourists."*

### 4.1.5. Openness to Experience

The experience of new things is usually considered as the motivation of some tourists in the tourism literature, while this study showed that escaping from routine and meeting new people had pushed some entrepreneurs towards the ecolodge business. A middle-aged woman who was the manager of a family ecolodge said:

*"When travelers from different areas of the country come to my ecolodge, my spirit is kept high".* *"The more crowded my ecolodge is, the more I enjoy it."* said the young man, who had set up an ecolodge with his family.

### 4.1.6. Availability of Loans

The availability low-interest loans to build or equip an ecolodge offered by the Ministry of Cultural Heritage, Crafts, and Tourism was among the drivers that influenced the decision of all interviewees to enter the ecolodge business. One of the owners said:

*"When we first went to the CHTO of Mazandaran, they said we would be given a loan. We were still assessing the probable benefits of an ecolodge business. When we saw that they were offering a low-interest loan, we were encouraged."*

Another interviewee stated:

> *"One of our relatives had come to us. He said that the CHTO of Mazandaran grants loans for traditional houses. He did not say what they were lending for. We said let's get this loan and spend it to complete the construction of our new house while repairing this traditional house. But when we went there, they said you must establish an ecolodge in your traditional house."*

We can consider drivers of unfavorable conditions for meeting some basic needs, poor working conditions, economic failure, openness to experience, and childhood influences as push factors which were related to the individual characteristics and conditions, while the receipt of a loan was a pull factor that came from the outside environment.

### 4.2. Entrepreneurs' Motives

Motives are a set of reasons for the behavior of individuals. The analysis of the data obtained from the interviews showed that the participants entered the ecolodge industry with different motives.

### 4.2.1. Making Money

Many ecolodge entrepreneurs spoke of financial motives when asked why they entered the business. Some of the interviewees started an ecolodge as their main source of income.

### 4.2.2. Employment Creation

Most interviewees cited the economic benefits for themselves in explaining the impacts of an ecolodge business, but some also mentioned job creation for local community members. Although few in number, these entrepreneurs felt compelled by a sense of social responsibility to do so. One of the interviewees, a teacher and her husband, who was a public servant, identified job creation for family members as one of the main reasons for entering the business. She had three brothers with low-paying jobs, and this put a lot of stress on her parents.

> *"I have three unemployed brothers. Really, they are not unemployed, but they work in the suburbs. I wanted them to have productive and decent employment here."*

A number of other interviewees took responsibility beyond their family and had a double commitment to local community and job creation for the villagers. These people generally believed that they would create jobs by employing local people or buying raw materials from the villagers. They identified job creation for villagers as one of the main motives for their entrepreneurship. For example:

> *"One of the things I wanted to do was create jobs. Now, every purchase I make or every guest I bring here benefits the people of this area."*

> *"I wanted to create a job for the people of this area. Tourists come here and buy dairy products. Its income benefits the villagers."*

### 4.2.3. Self-Reliance and Mastery

Earning independent income was another motive to set up an ecolodge. One of the young men interviewed, whose main occupation was animal husbandry, spoke of the motive for his financial independence:

> *"All my family members are ranchers, and we raise horses. But I wanted to have an independent income for myself."* Another interviewee described the motive for mastery and managing and leading other people:

> *"I enjoy managing here. Five people work for me. One cooks food. Someone is holding my baby."* Indeed, some of the entrepreneurs sought nonpecuniary benefits, such as being their own and others' boss.

### 4.2.4. Social Interaction

Affected by the openness to experience driver, a group of individuals referred to a set of motives expressing their social desires. For example, one of the interviewed women, in

response to the question, "Which one of your needs were met in this business", referred to the tendency to establish social relations as an important motive: *"I'm a social person. I enjoy having guests."* Explaining his interest in learning about diverse cultures, another interviewed entrepreneur said:

*"Every day, a new tourist a new culture; I get acquainted with a new culture. I ask my guests about their customs."* The last sentence implies the close interaction of the host with his guests. He stated as an enthusiastic entrepreneur: *"The entrepreneur who enters the ecolodge business with love and passion treats the tourist with patience and interest and with all his heart and soul"*.

### 4.2.5. Preservation of the Rural Lifestyle

A young interviewee with a university degree viewed the destination community's respect for the rural lifestyle as a key factor in his entry into the business:

*"I want people to respect our lifestyle. I am determined to show them how hard we work."*

The man believed that the rural community makes a great effort and bears the bulk of the burden of production. However, it does not receive the respect of the urban consumer community it deserves for the hard work. This causes the villagers to underestimate their lifestyle values and, as the importance of this lifestyle decreases, they lose their motive to produce and migrate to urban areas. Therefore, he was trying to restore the lost respect for this lifestyle by showing the importance and beauty of rural life to the residents of urban areas.

### 4.2.6. Protecting Cultural Heritage

A few entrepreneurs spoke about the need to preserve the area's native architecture. They perceived the preservation of traditional works as one of their motives for their entry into the ecolodge business. These interviewees believed that with the passing of time and the extensive demolition of old and traditional houses in rural areas of Mazandaran Province, very few of them remained and should be protected as the remains of their ancestors' lifestyles:

*"Our ancestors spent years trying to create this kind of architecture. Our job is to keep it."*

Another interviewee cited interest in his ancestral house and its gradual deterioration as one of his major motives in trying to repair it:

*"I loved this house so much. I was upset and depressed when I saw the house being destroyed. I always wanted to repair it."*

### 4.2.7. Protecting the Environment

Environmental motives were less pronounced, but some of the entrepreneurs' concerns about adverse environmental impacts, such as pollution of forests and the Caspian Sea coast, clearly emerged. Ignited by experts, an owner of an ecolodge with tertiary-level education had taken steps to protect the environment by buying some of the surrounding forest land to protect it:

*"Unfortunately, the people of Mazandaran do not know what the Hyrcanian forests are and where they grow. It is very important for me to preserve this forest."* He established his ecolodge with the help of his parents.

Another entrepreneur believed that the unplanned meeting of the needs of the urban dwellers for recreating in natural areas has caused severe damage to the agricultural lands of the Mazandaran Province by constructing second homes on these lands. Thus, by providing the right model of ecotourism, the needs of these people can be met, and agricultural lands can be protected:

*"If I serve urban people, they won't buy land here and build a villa."*

The combination of respect for the lifestyle of the villagers, self-reliance and mastery, preservation of cultural heritage, and protecting the environment are motives in line with

the results of McGehee and Kim [74], Nickerson, Black and McCool [47], and Phelan and Sharpley [49]. Contrary to Ollenburg and Buckley's [48] findings, economic motives play a strong role compared to other motives in persuading entrepreneurs to enter the ecolodge industry.

*4.3. Idea Sources*

The analysis of the interviewees' statements showed that the entrepreneurs of ecolodges had obtained the idea of establishing such a business from a variety of sources.

4.3.1. Experts at the CHTO of Mazandaran

Experts from the CHTO of Mazandaran were one of the most important resources that guided people interested in creating new ecolodge businesses. One of the interviewees was a person who sought to earn a living from tourism through his traditional house. He explained how the idea was formed:

> *"One of our families called his acquaintance at the CHTO of Mazandaran. We arranged an appointment for her to visit the house and give her opinion. For the first time, she introduced us to the idea of an ecolodge."*

Two other interviewees sought to set up a rural café and museum, while experts from the CHTO of Mazandaran had given them the idea of setting up an ecolodge:

> *"We wanted to establish a rural café. But when we went to CHTO of Mazandaran, they said that we have something called an ecolodge. You can get in this business."*

In almost all cases, CHTO experts played a significant role in persuading entrepreneurs. They assured individuals that the ecolodge entrepreneurship would give them a net profit. It is worth noting that government experts mainly played a role in encouraging people with traditional houses to convert them to an ecolodge establishment but not with respect to their training and counseling.

4.3.2. Advice of People Surrounding the Entrepreneur

Another source of information was people surrounding entrepreneurs, including relatives, friends, and acquaintances, whose advice was convincing. One of the entrepreneurs explained:

> *"One of our families came to our house. We had just come here. She [explain who this is] said why don't you rent this house to tourists? We said who's coming here? She showed a lot of photos that tourists in Yazd and Isfahan are staying in traditional houses. She said it has a good revenue. Then we went to the CHTO of Mazandaran [ . . . ]."*

4.3.3. Observing Successful Models

Another group of interviewees believed that neither of these sources played a role; rather, inspiration came from successful examples. For example, three interviewees said that they first observed the phenomenon of tourist accommodation in one of the villages around Tonekabon County arranged by a villager. This person had established an ecolodge and accommodated international guests. One interviewee said of him:

> *"Someone here has a house. His wife is Venezuelan. Now they bring guests and receive considerable income from these stays. They are rich. They don't accommodate Iranians."*

Furthermore, a young boy who started an ecolodge in the forest while preserving the environment was inspired by a pomegranate festival. According to the head of a village, villagers have gradually become more interested in the ecolodge business by observing the income generated in the village and its ability to provide for the household's livelihood due to the growing demand from tourists and, consequently, the number of entrepreneurs in this field increased. In fact, successful local models have given the entrepreneurs more confidence in setting up an ecolodge and its profitability.

### 4.4. Context

Based on interviews with ecolodge entrepreneurs, CHTO experts, and residents of the surrounding villages, we determined a set of conditions that provided the opportunity for entrepreneurs to enter the ecolodge industry and boost its pace of development.

### 4.4.1. Ownership of a Traditional House

Interviews revealed that according to the policies of the Iranian Ministry of CHTO, one of the conditions for granting permission to set up an ecolodge was the applicant's access to an old habitable house in rural and natural environments. In response to the question why only those with a traditional house can establish an ecolodge, one of the experts said:

*"Basically, the ecolodge project sought to preserve the region's traditional and indigenous architecture. For this reason, we do not grant licenses to those who want to build a modern ecolodge, because we are sure of the negative environmental and cultural effects of doing so. Since the launch of the ecolodge project in this province, about 80% of our requests have been for building a resort rather than enhancing the traditional houses and turning them into ecolodges."*

The CHTO has purely a legacy view of ecolodges and uses them as a strategy to preserve the traditional architecture of rural areas. Villagers surveyed also cited not owning a traditional house as one of the reasons they did not enter the ecolodge business:

*"We haven't a traditional adobe house. It is cemented. We have just built it. It is not suitable for this business."*

*"The tourist likes to see a mud and traditional house. Except these few houses [referring to old village houses], all have been destroyed."*

The ownership of a traditional house led to a drastic reduction in the cost of setting up an ecolodge, making this type of entrepreneurship achievable and affordable.

### 4.4.2. Having a Rural Background

The opinion of all interviewed entrepreneurs and experts was that an ecolodge is genuine when it is run by a local community member with an authentic indigenous lifestyle. They believed that an ecolodge run in a modern form and by non-indigenous people is fake and not worth it. One of the owners, who abandoned urban life after an economic downturn, said:

*"We do not playact. This is our life. If one comes from the city to start this, he has to pretend to be a villager. He can't. That's why it won't work."*

In contrast, there were others who were only interested in the village and said that they were willing to do the job of villagers and temporarily choose this lifestyle. However, the common feature of all interviewees was to have a rural background, whether they themselves, their parents, or their ancestors lived in the village.

### 4.4.3. High Potential of the Province as a Tourist Destination

The province of Mazandaran has beautiful natural scenery, cultural attractions, and agricultural products, which attract many domestic and international visitors every year. For example, over 106,000 foreign tourists visited Mazandaran Province during the first nine months of the Iranian year (March–December 2019). An interviewed village head believed that the diverse natural, cultural, and historical attractions in the area entice a large number of tourists to travel to this area. He said that the unmet demand of tourists for accommodation and hospitality services in the area has strengthened opportunity perception among the local community and encouraged them to become ecolodge entrepreneurs.

### 4.4.4. Family Companionship

All but one of the ecolodges in this study were family-run. One of the entrepreneurs described his wife's role in the establishment of the ecolodge:



*"We started the job with my wife. It is very important for me that she accepted to live in the village. [ . . . ] There must be a couple to run an ecolodge. [ . . . ] I couldn't buy a manteaux for my wife for 3 years. We spent everything we earned here. [ . . . ] Being a couple is very important to get started."*

Another interviewee described her husband's accompaniment:

*"My husband left the house his father had bought for us in the city, and he came here to start the job with me."*

A number of villagers who had a traditional house but had not established an ecolodge believed that they could not enter the business because of insufficient household labor.

According to the interviewees, in the early years of establishing an ecolodge, the profits are not sufficient to hire a worker. The relatively expensive restoration of the traditional houses prevents such costs. Access to family labor had reduced the perceived costs of setting up an ecolodge and vice versa.

*4.5. The Interaction Model of Ecolodge Entrepreneurship*

All of the components of drivers, motives, idea sources, and context are needed together to predict the likelihood of entrepreneurs' entry into the ecolodge business. For example, if an individual is exposed to a good idea but does not have the motive to do so or drivers are not sufficient to arouse their interests, one cannot expect an ecolodge entrepreneurial activity to take place. This is evidenced by villagers who did not enter this industry, while others had done so. Even though some local residents owned a traditional house, this was not sufficient to start an ecolodge business, as highlighted by a villager who cited his age, current job satisfaction, and lack of need for additional financial resources as reasons for not entering this industry. Without a traditional house, interviewed villagers found themselves unable to enter the ecolodge industry despite having strong motives and drivers. This was the case for eight of the interviewees, demonstrating the importance of the enabling context. Idea sources have also been effective in orienting motivated entrepreneurs who need new jobs toward tourism and convincing them to set up an ecolodge business. Some of the villagers were motivated to enter the ecolodge business but were not exposed to the idea of setting up an ecolodge. A rancher who owned a traditional house stated that *"I would prefer to make a living with my on-farm job and not enter a new business, because I am not familiar with the aspects of the new job"*.

Figure 1 depicts the interaction of drivers, motives, idea sources, and context in the evolution of an ecolodge entrepreneurship project. Therefore, each component appears in one or more of its variants and leads to ecolodge entrepreneurship. Putting some variants of these components together creates two types of entrepreneurs as follows:

1.  Initiating entrepreneurial activity with an economic motive, driven by an economic failure and/or availability of loans, receiving the idea of establishing an ecolodge from governmental experts, and relying on the use of a traditional house. This is the most common form of the entrepreneurial activity formation.
2.  Initiating entrepreneurial activity with non-economic motives, driven by openness to experience and childhood influences, receiving the idea of establishing an ecolodge by observing successful models, and relying on the rural background and family companionship. These motives include a mixture of social interaction, preservation of the rural lifestyle, job creation, and protection of cultural heritage and the environment. For example, pursuing childhood interests have pushed some entrepreneurs toward establishing an ecolodge to preserve the rural lifestyle.

In the next stage of the research, entrepreneurs from each of these two segments were asked about their ecolodge administration, to what extent the interaction between the tourist and the host is formed, and what kind of services were provided in the destination.

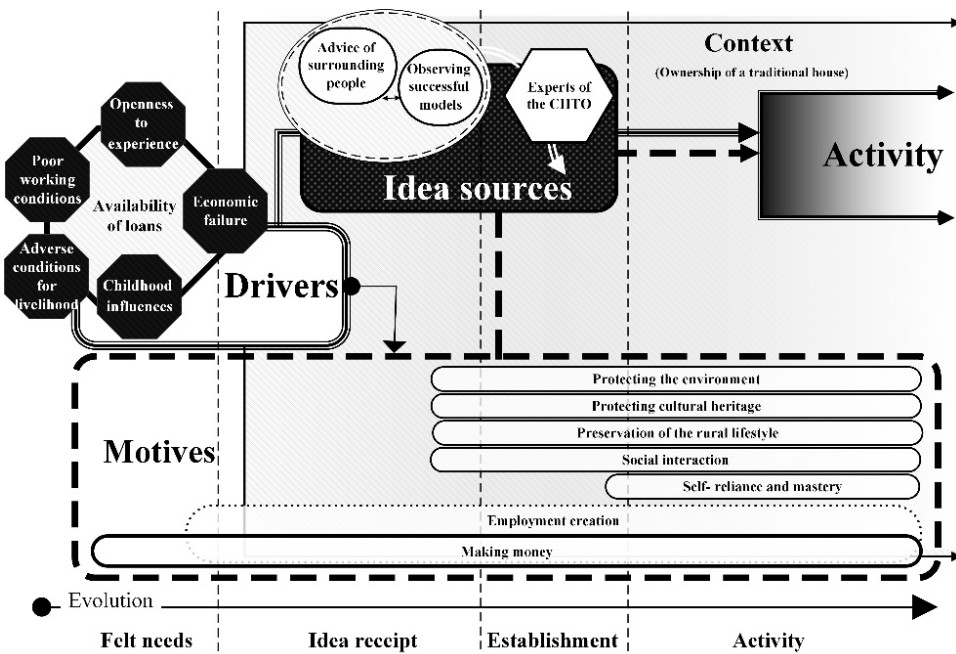

**Figure 1.** Interaction between drivers, motives, idea sources, and context in the process of entrepreneurs' entry into the ecolodge industry.

*4.6. Typology of Ecolodge Entrepreneurs*

Interview responses from the above two segments led us to a new classification of ecolodge entrepreneurs.

4.6.1. Ecolodge Lovers

Some of the entrepreneurs had a family-owned and -operated ecolodge business with two or more family members involved in its administration. The young people, along with the elderly members of the family, took on some of the responsibilities. For example, in the words of one these entrepreneurs: "*My mother is 54 years old; when I'm not at the ecolodge, she makes the necessary arrangements and accommodates the tourists. When a tourist comes, she bakes traditional bread and cooks Kalle-pache (sheep's head and trotters) early in the morning. Serving Kalle-pache with traditional bread is very enjoyable for tourists. My father has had a herd of sheep and has been raising chicken. My daughter is five-years-old and she helps pick up handicrafts in the ecolodge store. She wears local clothes and takes pictures with tourists.*"

A village head believed that "*the tourist prefers to talk to a local elderly person. An old knitter woman can share the memories of the village with the tourists while weaving. Older hosts are more patient and always reside in the local community and can guide the tourists.*"

Getting acquainted with different cultures, as well as preserving and strengthening the culture and environment of the local community, had a distinctly important and powerful role in their motivation. These entrepreneurs were self-motivated, not requiring external drivers to enter the ecolodge business. According to the findings of the interviews, this segment of entrepreneurs usually adhered to the following principals:

They preserved traditional architecture in their ecolodges, treated tourists honestly, and did not believe that money was everything. For example, a young married woman believed that the ecolodge entrepreneur should not enter tourism entrepreneurship just for the sake of getting more money: "*You should provide services to tourists with all your might. You should give more than your guest expects.*". Explaining how to provide honest services to tourists, she gave the following example: "*I once hosted a tour of 15 people, all of whom were women and retirees. I offered various services, such as accompanying them on the river and waterfall and brewing tea on a charcoal fire, but I only charged them for their overnight stay. They still call me, see the story of my Instagram page, and express their satisfaction.*".

These entrepreneurs were permanent residents of the local community and had set up an ecolodge along with their routine life. Therefore, starting this business had not led to a change in farmland use but provided services throughout the year. They had direct contact and good interaction with tourists and sought to create empathy with them. One of the entrepreneurs said: "*We try to encourage tourists to participate in some farming activities, such as feeding chickens, so that they do not feel strange and alienated.*" Hence, this segment was identified as ecolodge lovers.

### 4.6.2. Cool Job Seekers

Another type of ecolodge entrepreneur had entered the ecolodge business individually or as a family pulled by low-interest loans with the main motive of earning easy income. They were tired of their previous working conditions or were bankrupt in their previous jobs. An entrepreneur who was previously employed in construction said: "*Making money through ecolodge businesses is easy and fun.*"

They stayed in the local community mainly during good weather seasons, including spring and summer, and attracted tourists for 6 months of the year. In some cases, they did not even belong to the local community nor owned the ecolodge, in which case they rented the ecolodge from local community residents. This group of entrepreneurs had little knowledge of the culture and history of the local community. Their interaction with the tourist was limited to providing accommodation and food services and showing the attractions of the village. One of the experts from CHTO believed: "*strengthening the original culture of the local community is of little importance to this type of entrepreneur.*" Thus, this segment was labelled as cool job seekers.

### 4.6.3. Young Detached Entrepreneurs

Young people who had entered the ecolodge business as individuals and not as a family, and whose main goal was to earn an easy and cool income, did so influenced by external drivers such as low-interest loans offered by government. These entrepreneurs may not even be natives of the village, run the ecolodge by hiring labor, and usually live in the urban areas. They attracted tourists for a maximum of 6 months of the year.

Their interaction with tourists is minimal, and the business of the ecolodge has nothing to do with the routine and main life of the entrepreneur. These entrepreneurs often have negative cultural effects on the local community. They build new accommodation as an ecolodge and usually do not follow the ecolodge construction standards, leading to changes in farmland use. They only offer bed and breakfast services and basically do not promote or contribute to the local culture.

## 5. Discussion

According to the findings, self-reliance and mastery were found to be effective motives in ecolodge entrepreneurship. However, unlike Ainley and Kline's [38] assumption, but in line with Phelan and Sharpley [49], this study revealed that economic motives play key roles in ecolodge entrepreneurship. The findings counter the argument raised by Ainley and Kline [38], attributing the dominance of economic motives identified in previous studies to their positivist worldview. The classification of motives presented in this study can cover most of those revealed in previous research as an umbrella [45–50].

The in-depth analysis of interviewees' statements revealed that the process consists of four components, including drivers, motives, idea sources, and context. Together, these components made entry into the ecolodge business an accessible and profitable option for the entrepreneurs. We conclude that weighing costs and benefits of ecolodge entrepreneurship has been influenced by three types of forces, namely (1) causal conditions, (2) context, and (3) intervening conditions (Figure 2). The causal conditions include poor working conditions, unfavorable conditions for livelihood provision, economic failure, childhood influences, need for self-reliance and mastery, openness to experience, and a sense of social responsibility. Of these conditions, unsatisfactory job and living conditions

have reduced the entrepreneurs' expectations, scrupulosity, and threshold of the perceived negative impacts (perceived costs) and motivated them to earn more income by establishing an ecolodge. These conditions, in turn, can be due to the macro environment, resulting from factors such as decreased government support programs, income fluctuations in previous jobs, and climate change [38,39,42,43].

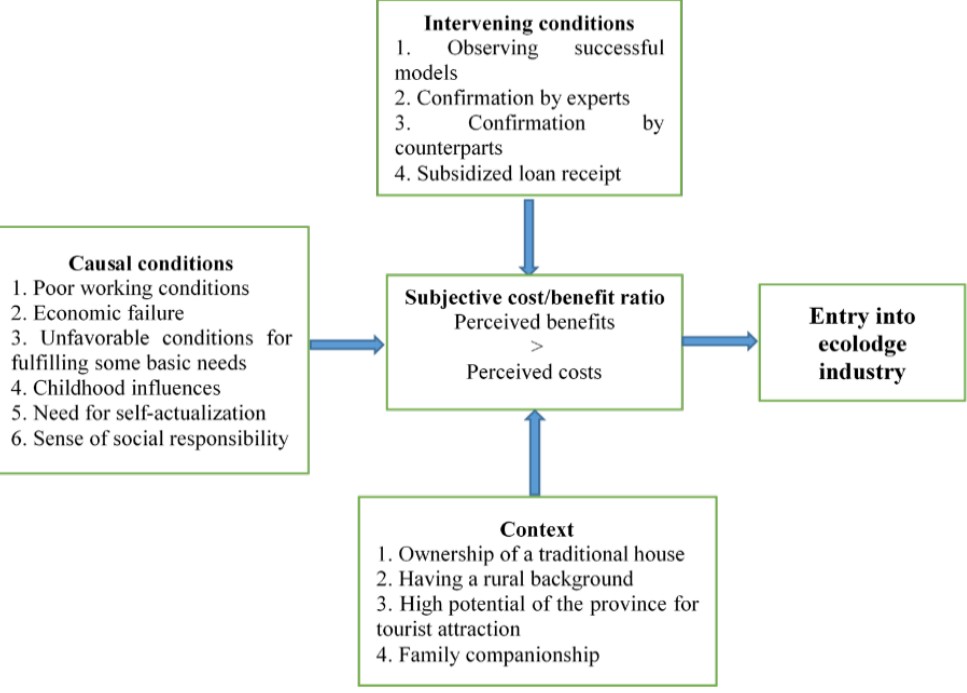

**Figure 2.** The process of ecolodge entrepreneurship.

Moreover, the context, including ownership of a traditional house, having a rural background, and so forth, alongside intervening conditions, such as the availability of subsidized loans, observation of successful models, and persuasion by experts and counterparts, have led to the entrepreneurs' enhanced perception of the positive impacts (perceived benefits) and reinforced their motives. Similarly, the context and intervening conditions have led to more resources and fewer perceived obstacles and, consequently, resulted in better perceived behavioral control by the entrepreneurs and their assurance of successful entry into the ecolodge business. In addition, the "need for self-actualization", openness to experience and a sense of responsibility towards the unemployed youth of the family reinforced perceived benefits of the ecolodge entrepreneurship compared to its perceived costs. Ultimately, the entrepreneurs felt that the potential benefits outweighed the potential costs and, therefore, decided to enter the new business (Figure 2).

In this study, we developed a new typology of ecolodge entrepreneurs, which captures their heterogeneity in terms of their drivers, motives, idea sources, and context, as well as their business characteristics and administration. Compared to the conventional binary classification [53–60], we have identified three homogenous types of ecolodge entrepreneurs, including *ecolodge lovers*, *cool job seekers*, *and young detached entrepreneurs* (Figure 3). The ecolodge lovers are intrinsically motivated and establish the business mainly for the sake of getting acquainted with different cultures, job creation for the villagers, and preserving and strengthening the culture and environment of the local community. Such motives lead this group of entrepreneurs to adhere to the principles of sustainable tourism and good interaction with tourists. Because these entrepreneurs have lived in the local community for generations, they are more likely to be attached to the place and create better cultural and environmental impacts. The embeddedness of the ecolodge business in the local community is high. When the entrepreneur, along with their family, is a member of the local community and attracts tourists while maintaining their routine activities, such

as beekeeping, animal husbandry, and crop production, then they are more prepared to extend hospitality at any time of the year. Thus, these businesses are more likely to be year-round ecolodges. For example, in off-seasons, such as winter, this type of entrepreneur can accommodate tourists even with a small number of tourists, because they save on overhead costs, make the most of their routine facilities for hospitality, and value creation and require minimal inputs and external resources.

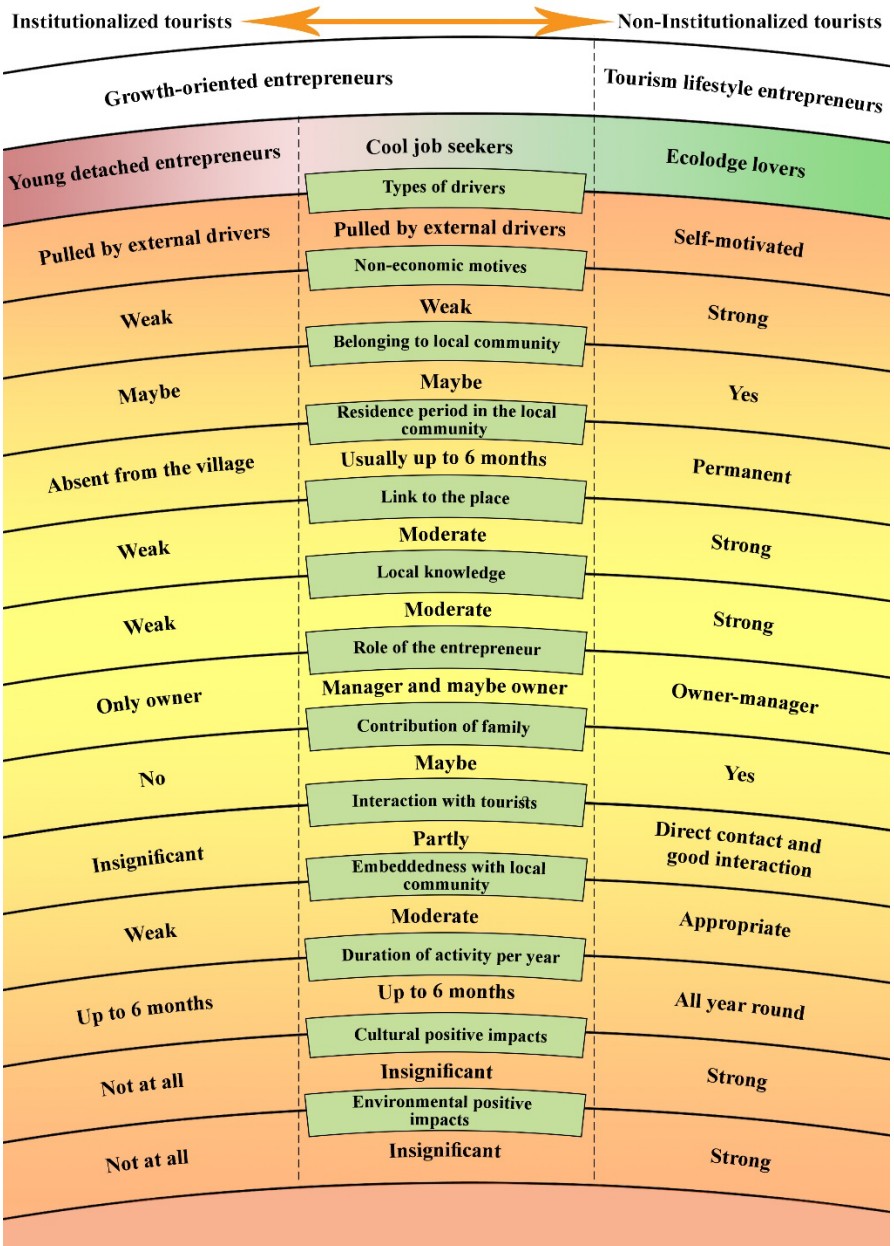

**Figure 3.** Three clusters of ecolodge entrepreneurs.

Tourists are attracted to older men and women, as they tend to wear traditional clothes, tell stories and narratives, and cook traditional foods. They have a strong place attachment and extensive local knowledge. The importance of local knowledge, including narratives, has been recognized by Dias et al. [56] and Shrivastava and Kennelly [69]. However, older people usually do not innovate well. Therefore, the collaboration of the elderly and young people is one of the determinants of the distinctive performance and competitiveness of the ecolodge lovers. Ecolodge lovers have some of the characteristics of tourism lifestyle entrepreneurs, such as good innovation, strong non-economic motives,

good local knowledge, and high place attachment. However, contrary to Fu et al.'s [57] statement that tourism lifestyle entrepreneurs may migrate to a destination to start a tourism business, ecolodge lovers have usually lived in the local community for multiple generations and have deep local knowledge, which allows them to create unique and differentiated opportunities for tourists. Despite their differences, the segments of cool job seekers and young detached entrepreneurs were pulled by subsidized loans, and the economic benefits of the tourism business are a priority for these entrepreneurs (Figure 3).

## 6. Conclusions

For the first time this study examined the systematic process individuals engage in to enter the ecolodge industry in rural areas in an emerging market, where there are government incentives to do so. The ecolodge entrepreneurs were motivated by different motives, including making money, employment creation, self-reliance and mastery, social interaction, preservation of the rural lifestyle, protecting cultural heritage, and protecting the environment, although the economic motives were more pronounced. The ecolodge entrepreneurship process consists of four components, including drivers, motives, idea sources, and context. This led us to propose a new conceptual framework to explain how these components formed the overall positive perception of entrepreneurs and finally induced them to set up an ecolodge. Three types of forces, namely causal conditions, context, and intervening conditions, influenced perceived impacts of ecolodge entrepreneurship.

By adopting a qualitative and in-depth approach, the present study developed the new classification of tourism entrepreneurs. Based on the entrepreneurship process and ecolodge administration and services, this study identified three types of ecolodge entrepreneurs, namely ecolodge lovers, cool job seekers, and young detached entrepreneurs. Our typology provides more details about the entrepreneurs' characteristics. The segment of ecolodge lovers is the most susceptible one to the establishment of an authentic ecolodge businesses. Ecolodge lovers are most interested in interacting directly with tourists, a core aspect of ecotourism. Characteristics such as strong non-economic motives, good local knowledge, high place attachment, long stay throughout the year in the local community, and administration of the business through the participation of family members have made these entrepreneurs the most compatible segment with the authentic ecolodge development. Compared to the other two segments, ecolodge lovers can better satisfy non-institutionalized tourists, especially drifters [75], because of this interest in interacting directly with tourists. Nevertheless, the findings of this study showed that the majority of entrepreneurs belong to the other two segments, who have entered the ecolodge industry mainly with the aim of achieving an easy income, regardless of ecolodge requirements and rules. At no time were the professional competencies and insights required to manage an authentic ecolodge considered. We conceptualized this process of entering the business as "*democratized unplanned (haphazard) ecolodge entrepreneurship in an emerging market*". This is where individuals establish pseudo-ecolodges based on traditional business wisdom [10,17] without sufficient tourism knowledge and experience [23]. The explored process confirms the concerns of international scholars with developing ecolodges in an unsustainable manner. This process is rooted in the approach that considers an ecolodge business as a semi-skilled job and a tool for rapid job and income creation in rural areas. Here, the main motive is to earn money, reinforced by drivers of job dissatisfaction and poor living conditions. Ownership of a traditional house is considered as the main criterion by the government.

The findings are critical of the current policy framework of ecolodge development in a country such as Iran, and they illuminate the need for its review and revision. This study enables the formulation of instructions for planning schemes to promote authenticity of ecolodges in emerging markets. Developmental organizations, such as non-governmental and non-profit organizations, must plan to empower local entrepreneurs in networking, acquiring technical competencies, and environmental awareness. Establishing and developing the labeling schemes and promoting the viewpoints of citizens about the principles of

ecotourism and ecolodge entrepreneurship can play an effective role in the development of authentic ecolodges in emerging markets. It seems that measures have recently been taken to improve the existing policies, especially through non-governmental organizations. Moreover, the process of ecolodge entrepreneurship and typology of ecolodge entrepreneurs explored in this study can be the basis for future quantitative research. Future studies should quantitatively measure the variables of causal and intervening conditions and then assess the causal relations between them with the perceived impacts of ecolodge entrepreneurship and, finally, the intention to enter the business. The contextual variables can also be studied and assessed as moderators in the structural equation model. The process explored in this research was documented in a region that is considered as an ecotourism hub and one of the most important tourist attractions in Iran. Furthermore, in Mazandaran, the ecolodge industry is officially only six-years-old, and there is a lot of unmet demand for accommodation and hospitality services.

However, conducting a similar study in areas that are typical in terms of the attractions and visitors could lead to different findings, even in emerging countries. In such areas, the impact of natural attractions would probably be decreased, and the importance of skills and expertise seems to be increased. In fact, in such situations, individuals are likely to see the ecolodge industry as a skill-intensive business. This could lead to its planned development, according to the philosophy and principals of ecotourism. Similar results are expected in future studies in areas and countries with a long history of ecolodge development and with highly competitive markets and experienced ecotourists. Future research should explore whether the process of entrepreneurs' entrance into the ecolodge business in these countries has different components and stages and, if so, whether this difference leads to different outcomes. Moreover, this cross-sectional research study is not able to examine the possible changes in the motives of the entrepreneurs for the continuation of the business. Therefore, longitudinal studies should be undertaken to investigate the possible changes. With more experience, will the motive of entrepreneurs evolve and move towards real ecotourism? If yes, how?

**Author Contributions:** Conceptualization, H.V.; Data curation, S.H.M.; Methodology, H.V.; Project administration, H.V. and K.K.; Supervision, K.K, M.J. and A.D.; Writing—original draft, H.V. and S.H.M.; Writing—review & editing, M.J. All authors have read and agreed to the published version of the manuscript.

**Funding:** This research received no external funding.

**Institutional Review Board Statement:** Not applicable.

**Informed Consent Statement:** Not applicable.

**Data Availability Statement:** The study did not report any data.

**Conflicts of Interest:** The authors declare no conflict of interest.

## Appendix A. Entrepreneurs' Entry into the Ecolodge Industry

The process of entrepreneurs' entry into the ecolodge industry has been neglected. Previous work in turn has been carried out under five themes:

1) unsustainable ecotourism business practices [23,76,77],
2) profile of ecotourists, their motivations, characteristics, attitudes, satisfaction and related segmentation [39,78–90],
3) service quality, attributes, performance goals and sustainable management of ecolodges [15,18,91–94]
4) contribution of ecolodges in local sustainable development [95–97], and
5) miscellaneous themes namely private tourism ecolodge concessions, pedagogical process for designing an ecolodge [98,99].

**Table A1.** The ecolodge empirical studies.

| Type | Author (s) | Main Objective | Sample | Analysis | Key Findings |
|---|---|---|---|---|---|
| **Unsustainable ecotourism business practices** | Arze and Holladay [23] | Understanding the basic causes of unsustainable ecotourism business practices | Yacuma River Protected Area, Bolivia | Field note | Government is mostly focused on economic impacts of ecotourism businesses which are likely lose its market share due to environmental degradation. |
| | Fennell and Markwell [76] | Understanding ethical and sustainability dimensions of foodservice in ecotourism businesses | 84 accredited Australian ecotourism businesses | Manifest content analysis | Very few ecotourism businesses have been mentioned ethical and sustainability dimensions of food on their websites. |
| | Higgins-Desbiolles [77] | Exploring nature of trade-offs between environmental conservation and greater economic growth through tourism development | 26 focused interviews with stakeholders of an ecolodge on Kangaroo Island, Australia | Case study analysis | Economic development has been promoted at the expense of the environment. |
| **Characteristics of ecotourists** | Ban and Ramsaran [78] | Exploring service quality attributes of ecolodges | Interviews with 25 tourists | Qualitative coding and labeling | Compared to SERVPERF instrument, three additional dimensions were specific to the ecolodge sector. |
| | Beaumont [79] | Investigating Pro-environmental attitudes among ecotourists | 243 domestic and international visitors of Lamington National Park, Australia | Quantitatively examining differences between visitors | No significant differences in pro-environmental attitudes between ecotourists and non-ecotourists. |
| | Brochado and Pereira [80] | Understanding consumers' perceptions of services provided by glamping facilities | 172 comments rating glamping sites; 166 visitors staying in Portuguese glamping facilities | Content analysis and Exploratory factor analysis | Service quality in glamping was multidimensional, including five facets: staff, tangibles, food, nature-based experiences, and activities. |
| | Chan and Baum [81] | Exploring ecotourists' perceptions of ecotourism experiences | Interviews with 29 European ecotourists staying at 2 ecolodges in Sukau, Lower Kinabatangan, Malaysia. | Qualitative analysis techniques | The preferred ecotourism activities included ecotourists' physically engaging; interacting with the site service staff; socialising with other ecotourists, and acquiring information during the visit. |
| | Chan and Baum [39] | Exploring ecotourists' motivation factors in the ecolodge accommodation | Interviews with 29 ecotourists staying in 2 ecolodges, Sukau, Malaysia | Qualitative phenomenological approach | Ecotourists were primarily attracted by the pull factors (destination attributes). |
| | Kwan, Eagles and Gebhardt [82] | Determining the characteristics and travel motivations of ecolodge patrons | 331 ecolodge patrons at 6 ecolodges, Cayo District, Belize | Chi-squared test or an ANOVA test | There were significant differences found in travel motivations and the importance of ecolodge attributes amongst the price levels. |
| | Kwan, Eagles and Gebhardt [83] | Determining the demographic and trip characteristics, and travel motivations of ecolodge patrons | 331 tourists who stayed at ecolodges, Cayo District of Belize | Importance – performance analysis | Ecolodge patrons were typically highly educated with high annual household income. Performance scores exceeded importance scores for all of ecolodge attributes. |
| | Lawton [84] | Profiling a sample of older adult ecotourist | 1140 ecolodge patrons, Lamington National Park, Australia | Chi-square or t tests | The older adult ecotourists preferred a higher level of comfort and less risk in comparison with the younger ecotourists. |

**Table A1.** *Cont.*

| Type | Author (s) | Main Objective | Sample | Analysis | Key Findings |
|---|---|---|---|---|---|
| **Characteristics of ecotourists** | Lu and Stepchenkov [85] | Classifying satisfaction attributes with ecolodge stays | 373 reviews extracted from TripAdvisor | Content analysis and a two-step statistical procedure | Twenty-six ecotourists' satisfaction attributes were identified and were classified into four groups: criticals, satisfiers, dissatisfiers, and neutrals. |
| | Mafi, Pratt and Trupp [86] | Exploring underlying satisfaction attributes of ecolodges | 11 guests in Matava ecolodge, island Kadavu, Fiji | Thematic analysis technique | There were emerging attributes that relate to local culture, local cuisine, local people engagement and remoteness. |
| | Newsome, Rodger, Pearce and Chan [87] | Understanding motivations and satisfaction of tourists with wildlife tourism experience | 346 visitors of eight lodges along the Kinabatangan River, Malaysia | Importance-performance matrix | The tourists, despite their satisfaction, were concerned about the number of boats and the protection of the River. |
| | Simpson et al., [88] | Identifying factors influencing revisit intention of ecolodge visitors | 362 visitors to Sri Lankan ecolodges | Structural modelling | Travel motives and satisfaction had a significant impact on tourist intentions to revisit individual ecolodges. |
| | Weaver and Lawton [89] | Segmenting overnight ecotourist market | 1800 tourists of ecolodges, Lamington National Park, Australia | Cluster analysis | The ecotourists were segmented into "harder", "softer" and "structured" clusters. |
| | Weaver [90] | Identifying the characteristics of hard-core element within a special population of ecotourists | 1800 ecolodge patrons from two ecolodges, Lamington National Park, Australia | Cluster analysis | The hard-core ecotourists fitted in the expectations of the hard ecotourism ideal type. |
| **Conditions, performance and goals of ecolodges** | de Grosbois and Fennell [91] | Identifying best practices in the sustainable management of the world leading ecolodges | 65 top global ecolodges | Qualitative content analysis | Key themes included biodiversity conservation, preserving socio-cultural heritage, improving social wellbeing, learning opportunities for guests, etc. |
| | Hu et al. [92] | Exploring green attributes of an ecolodge | Triangulated sources of the organization's publicity materials, online comments and observation from the Crosswaters Ecolodge and SPA, China | A multi-method qualitative study based on grounded theory | There were four green attributes: functional values, aesthetic values, psychological appeal, and experiential appeal. |
| | Lai and Shafer [15] | Exploring how ecotourism is marketed through the Internet | 35 ecolodge operators from 14 Latin America and the Caribbean | Content analysis | Most of the online marketing messages sent by the ecolodges only partially aligned with ecotourism principles. |
| | Mic and Eagles [93] | Investigating development of a cooperative brand strategy for midscale ecolodge businesses | 12 ecolodge owners and managers in Costa Rica | Thematic analysis | Individual operation of most ecolodges has imposed higher marketing and management costs. Development of a cooperative brand had challenges in 8 areas. |
| | Osland and Mackoy [18] | Discovering ecolodges' performance goals and assessing their performance | Interviews with owners and managers of 21 ecolodges in Costa Rica and Mexico | Qualitative analysis techniques | A total of 84 performance goals were identified and classified using a new framework. |

**Table A1.** *Cont.*

| Type | Author (s) | Main Objective | Sample | Analysis | Key Findings |
|---|---|---|---|---|---|
| | Suarez-Dominguez, Argudo-Guevara and Arce-Bastidas [94] | Identifying important characteristics of ecolodge | Tourists who visited the Ecolodge Napo Wildlife Center, Ecuador | Content analysis | The location was one of the most important attributes of the ecolodge. |
| **Contribution of ecolodges in sustainable development** | Hunta, Durham, Driscoll and Honey [95] | Investigating the contribution of ecotourism in local sustainable development | Interviews with 128 local residents, incl. 70 ecolodge employees, Costa Rica's Osa, Peninsula | Quantitative analysis/qualitative coding by thematic content | Ecotourism had most important contribution in improvements of residents' quality of life amongst locally available economic sectors. |
| | Keough [96] | Exploring compliance of the Wenhai ecolodge with the principles of ecotourism | N/A | Case study | The compliance of the ecolodge with the principles of ecotourism was confirmed. |
| | Little and Blau [97] | Exploring multifaceted benefits and effects of agritourism | 21 stakeholders in the four eco-lodges, Mastatal, Costa Rica | Mixed method approach | Agritourism was an effective adaptation strategy to climate change and economic stressors. |
| **Miscellaneous themes** | Coghlan and Castley [98] | Identifying perceptions of private tourism ecolodge concessions by residents, regulars and locals | 314 domestic visitors of Kruger National Park, South Africa | Descriptive statistics and content analysis | Those who lived closer to the park were significantly more likely to have more experience and better knowledge of the concessions but were also less likely to support their existence than other respondents. |
| | Gawad [99] | Presenting an example of a design studio pedagogical process for designing an ecolodge | Feedback from 210 students on overall learning experience of designing an ecolodge in 2 touristic locations in Egypt. | Descriptive statistics | The teaching process of students' ecolodge projects may include other supplementary components including invited guest speakers, field trips and so on. |

Note: N/A means not applicable.

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
