# Peer review of "Ecolodge Entrepreneurship in Emerging Markets: A New Typology of Entrepreneurs; The Case of IRAN"

_sustainability, doi:10.3390/su14148479_

Round 1
Reviewer 1 Report
The authors make a critical analysis of tourism businesses focused on environmental sustainability
Reference is made to specialized works from long periods of time (a.i. Duffy, 2006; Mehta, 2007 etc) commenting on recent developments in the business sector, ecolodge businesses, in emerging markets. Given the dynamics of such business in recent decades, as well as the strategic orientations and policy recommendations of international organizations, as well as the targets assumed by countries for the SDGs, I consider it necessary for the authors to present the critical analysis of the sector through stages of evolution, with a selection and analysis of good practices or examples of "formalism" (bad practices).
It is necessary to differentiate the content of the concepts used from the beginning - ecolodges, ecotourism business, etc.
Also, make the necessary clarifications to justify why, the references to the same country (Iran) mention once "Iran as a transitional country" and immediately in the following paragraphs "In emerging markets such as Iran".
“Citation of citation” system it is not recommended in scientific papers. Literature analysis involves in-depth knowledge of primary sources (the original works referred to) and critical analysis from the authors' perspective, and not taking opinions from other works ("Mohammadi, 2010 as cited in Alipour Eshliki and Kaboudi, 2012"; "de Haas, 2002; Scheyvens, 1999, as cited in Wondirad, 2019") I recommend giving them up or justify why such indirect comments are considered, which may alter the original message of the authors.
It is necessary to reformulate research question 4 "Can a typology of the entrepreneurs be developed?" with clear specification of the perspective / classification criterion.
How is the "ecolodge" sector and "ecotourism sector" statistically defined in the NACE Rev 2 classification?
2.1 Further clarification is needed on the use in the business environment and the addressability of policies for the ecolodge sector, at least to justify the application of qualitative research
2.2. What method of querying the 2 databases (WoS and Scopus) was used given the degree of overlap between them? What method of processing the selected sample was used? What method of differentiating the 4 groups was used? An adequate, scientifically substantiated presentation is needed, justifying the opportunity and fairness of the selection of literature analysis methods.
3.1. The sampling methodology and the criteria tree for defining the sample representativeness are not clearly presented.
3.2. What is the number of validated questionnaires and what is the degree of representativeness of the results, the added value for the typology approach proposed by the authors? What are the limitations of the methodology used? What demographic criteria of the interviewees were taken into account?
4.1. You mention "Five drivers" and then you detail 6 (4.1.1.-4.1.6)!
4.1.1. What does "educated" interviewee mean? It is necessary to indicate the level of the graduated school - primary, secondary or tertiary level of education or similar, according to international classification (for example ISCED)
At 4.1.- 4.4. a statistical analysis of the questionnaires is necessary and not only the exemplification of some answers to the open-ended questions.
4.4.3. In order to define the tourist attractiveness, it is more appropriate to use a statistical indicator relatively associated with the absolute one mentioned, in order to identify the comparative advantage of the selected area and / or the increased interest of tourists, over time.
Comments from 4.5 are limiting given the openness of the last decade to promote both tangible and intangible heritage for rural tourism development, considering that intangible heritage is also a consistent driver for ecolodge entrepreneurship development.
4.6. What is the scientific typology criterion used?
5. The conclusions need to be reorganized. The information in this section, including the figures, needs to be repositioned - in Chapter 4 and clearly specify the originality, relevance and scientific robustness of the results. It is necessary to refer to the research questions and the relevance of the results
Additional comments
Communicate your unique point of view to stand out. You may be building on a concept already in existence, but you still need to have something new to say. Make sure you say it convincingly, and fully understand and reference what has gone before.
Literature review needs substantial improvement and update since it is descriptive, and its purpose should be to critically evaluate relevant theories and models conceptualizing the topic. What did the authors find from the literature that a few other recent literature reviews on the subject have not noticed?
The methodological section also needs substantial improvement making explicit key methodological elements and the justification of the relevance and scientific value added of the selected research method.
The evaluator did not have at his disposal the Annexes indicated by the authors and therefore we cannot make detailed assessments on the robustness of "state of the art" and the fundamentals of the development of qualitative research.
Reviewer 2 Report
Thank you for the opportunity to review this manuscript. The topic is interesting and worth exploring. However, there are a few shortcomings worthy of consideration and inclusion in the revised manuscript which are pointed out in the comments below.
Comment 1: The introduction is well written, but you should emphasise more what is of added value and at the end of this section please add the structure of the work. In addition, the title mentions emerging markets, yet the research questions and the purpose of the paper omit this.
Comment 2: Due to the fact that the authors conduct the research only for Iran, so the title of the paper should be changed to: "Ecolodge entrepreneurship in Iran: a new typology of entrepreneurs".
Comment 3: I doubt that on the basis of 31 interviews the results can be generalised to the whole population. Besides, was the survey representative?
Comment 4: The results indicated in Figures 3 and 4 should be in Section 4. The Conclusions section should bullet the main findings. Also, as there are elements of discussion in this section, there should either be 2 sections: Discussion and Conclusion or rename this section to "Discussion and Conclusion".
Comment 2:
Reviewer 3 Report
The topic of the paper is interesting, I congratulate the group of authors:
The main suggestions for changes are as follows:
Please follow the template for bibliographic citations in the text they are numbered in square brackets.
I recommend, after presenting the research questions, the elaboration of the research hypotheses.
I consider that there are too many sub-sections, and it is lost from the cursive nature of the paper. I recommend deleting the 3rd order subsections (eg 4.1.1, 4.1.2. And so on).
The conclusions section should be very concise and the key information should be presented. I recommend dividing this section of conclusions into two parts: one for discussion (in which the figures will be presented) and the second for the main conclusions section.
Follow the template and the list of bibliographic references.
Round 2
Reviewer 1 Report
No additional comments
Author Response
Dear editor
The first respected reviewer has not made any new comments.
Best regards,
Hojjat

Reviewer 2 Report
Thank you for the opportunity to review this article again. I believe that the current version of the revised manuscript is worth accepting. In general, the text is now well written. I have only 1 comment regarding the appendices, specifically the text under Table 1. Putting this in its current form makes it not quite clear what the text refers to and causes confusion for the reader.
Author Response
Dear reviewer
Thanks a lot for giving us opportunity for revision of the paper again.
The sentence below the table was moved to the beginning and after the following sentence:
"Various initiatives have been launched globally to ensure the quality and compliance of tourism businesses with the principles of ecotourism. In the table (1), 13 initiatives have been reviewed". Please see P. 21.
Best regards,
